# Associations between humiliation, shame, self-harm and suicidal behaviours among adolescents and young adults: A systematic review protocol

**Aoibheann McLoughlin[1]ᵒ\*, Anvar Sadath[2]ᵒ\*, Elaine McMahon[2]‡, Katerina Kavalidou[3]‡, Kevin Malone[1]‡**

**1** Department of Psychiatry and Mental Health Research, St. Vincent's University Hospital, University College Dublin, Dublin, Ireland, **2** School of Public Health and National Suicide Research Foundation, University College Cork, Cork, Ireland, **3** National Clinical Programme, HSE, Dublin, Ireland

ᵒ These authors contributed equally to this work.
‡ These authors also contributed equally to this work.
* aoibheannmcloughlin@gmail.com (AM); anvarvakkayil@gmail.com (AS)

**Data Availability Statement:** No datasets were generated or analysed during the current study. All

## Abstract

### Background

Suicide is the second leading cause of death among young people worldwide and remains a major public health concern. Research indicates that negative social contexts involving familial and peer relationships, have far-reaching influences on levels of suicidal behaviours in later life. Previous systematic reviews have focused on evaluating associations between negative life events such as abuse and bullying in childhood and subsequent self-harm or suicidality. However, the association between adolescent experiences of humiliation and shame, and subsequent self-harm or suicidal behaviour among children and young adults has not been well examined. As such, this systematic review is conducted to examine the prevalence and association between humiliation and shame and self-harm, suicidal ideation, and death by suicide among adolescents and young adults.

### Methods

A systematic literature search in extant electronic databases including; MEDLINE, Web of Science Core Collection, CINAHL, PsycINFO, and Embase will be conducted to identify potential studies. Google Scholar, and the reference list of the retrieved articles and/or previous systematic reviews in this area, will also be scanned to identify further potential studies. ProQuest will be searched to identify relevant studies available within grey literature. There are no restrictions on the date of publications. Based on our initial review, the following terms were identified: *Population*: Adolescent (MESH), young adult (MESH), teen, teenage. *Exposure*: Humiliation, degradation, shame (MESH) or embarrassment (MESH), harassment victimisation, abasement. *Outcome*: Self-injurious behaviour (MESH), suicide (MESH), suicide attempted (MESH), suicide completed (MESH), self-harm, intentional self-injury, deliberate self-harm, overdose, deliberate self-poisoning, non-suicidal self-injury,

relevant data from this study will be made available upon study completion.

**Funding:** This study is being conducted as part of a funded collaborative study grant awarded to McLoughlin A by the Health Service Executive (HSE) - National Office for Suicide Prevention (NOSP), Ireland. The funders had and will not have a role in study design, data collection and analysis, decision to publish, or preparation of the manuscript. https://www.hse.ie/eng/services/list/4/mental-health-services/nosp/.

**Competing interests:** The authors have declared that no competing interests exist.

self-mutilation, suicidal thought, suicidal ideation, suicidal intent, suicide. At least one term from each category will be used for conducting the literature search. All original quantitative studies published in the English language which examined the prevalence or association between humiliation or shame and self-harm and/or suicidal ideation and/or completed suicide will be included. The studies will be assessed for methodological quality using the Joanna Briggs Institute critical appraisal tools. Narrative synthesis will be performed for all of the studies. If the studies are sufficiently homogenous, the results will be pooled for a meta-analysis. This systematic review protocol followed the Preferred Reporting Items for Systematic Reviews and Meta-Analysis Protocol (PRISMA-P) guidelines. The protocol has been registered with the International Prospective Register of Systematic Reviews (PROSPERO) [CRD42022289843].

## Discussion

This is the first review to synthesise evidence on the prevalence of, and associations between the experiences of humiliation and shame and subsequent self-harm and/or suicidal behaviours among adolescents and young adults. As there is growing evidence on increased self-harm among this age group, it is important to identify population-specific risk factors for self-harm and suicidality which will have significance in formulating tailored and effective treatment and therapeutic services for adolescents and young adults.

## Introduction

Self-harm and suicide remain a major public health concern for adolescents and young adults across the world [1,2]. Suicide is the second leading cause of death among young people worldwide, with several countries reporting increases in self-harm among this cohort in recent years [3,4]. Globally, suicide is the leading cause of death for female adolescents and the third most common for male adolescents after road traffic accidents and violence [5]. Suicide and suicidal behaviour become increasingly common after puberty. This is a trend likely attributable to new-onset mood disorders and substance abuse [6], school/family problems, conflictual peer relations, adverse early childhood experiences [6], and evolving personality factors including neuroticism and impulsivity [7]. Cognitive immaturity, lack of judgment and low impulse control play an important role in this increased risk of suicide [6], with research demonstrating an association between adolescent neurobiological changes and increased risk-taking behaviours [8]. Life stress is a critical factor in all major theories of suicide [9,10]. Life stressors, including acute life events, chronic difficulties, and trauma are associated with both suicidal ideation and attempts in adolescence and adults [9,11]. Evidence for an association between negative life events and suicidality is consistent [9,11]. Many adverse experiences resulting from peer relationships, peer conflict, victimisation and isolation are associated with suicidal behaviours (4, 12), with recent longitudinal research revealing that key social contexts in early adolescence, (involving familial and peer relationships), have far-reaching influences on levels of suicidal behaviours in later life [12].

The integrated motivational-volitional (IMV) model [13] is a prominent and widely accepted theoretical conceptualisation [14] that explains suicidal behaviour as a process made up of three distinctive phases in which different mediating factors are involved. In the *pre-motivational phase*, biological, and/or genetic and cognitive susceptibility factors pre-dispose

the individual towards suicidal behaviour. For instance, decreased serotonergic neuro-transmission (biological vulnerability) and/or socially prescribed perfectionism (cognitive vulnerability) are factors implicated in pre-disposing an individual to self-harm or suicidal ideation [13]. According to this model, socially prescribed perfectionism can intensify the feeling of defeat when an interpersonal crisis occurs [13]. Higher levels of perfectionism are also associated with sensitivity to emotional pain [14]. In the *motivational phase*, suicidal ideas and plans begin to develop due to the influence of negative feelings such as defeat and humiliation. When a vulnerable individual appraises that there is no perceived escape from humiliation or defeat, a sense of entrapment develops, a phenomenon which is a proximal predictor of suicidal ideation [13]. However, the progression of entrapment into suicidal ideation is dependent on the presence of motivational moderators. Protective motivational moderators in the form of social support, belongingness, and reasons for living allow the trapped individual to see positive alternatives, while feelings of burdensomeness, poor social support and depleted resilience will increase the risk of entrapment transitioning into suicidal ideation/intent [13]. Finally, in the *volitional phase*, precipitating factors such as access to means or impulsivity lead to a suicide attempt [13].

The role of humiliation/negative appraisal leading to a sense of inescapable entrapment in vulnerable individuals is a core feature of progression towards suicidality in the IMV model. In the context of negative life events, humiliation is the single most experienced life stress among adolescents followed by interpersonal loss [9], and thus warrants further attention. The process of 'humiliation' refers to two different forms of experience. Firstly; the act of humiliating or being humiliated, and secondly; the state or feeling of being humiliated. Thus, humiliation can be considered as an external event or an internal state. For the purpose of this systematic review, humiliation is conceptualised as an internal state from the recipient's point of view. Here, humiliation encompasses the intrinsic experience of being rejected, excluded, and put down [9].

Humiliation is associated with many mental health conditions. Persistent fear of being humiliated or scrutinised by others are common in social anxiety disorder [15], while suffering severe public humiliation can lead to major depression [16,17], hopelessness, and helplessness [18], and is associated with suicidal ideation or acts [19]. Moreover, past interpersonal humiliation events have been found to predict a higher level of persecutory ideation in a non-clinical population [20].

Whilst there is growing research on the topic of bullying [21–28], which can be inferred to be a cause of humiliation given threats to the self [29], bullying constitutes an event that is inflicted on a person, and does not capture the emotional reaction of the victim in the face of such harassment or abuse [29].

Shame can be understood as a cognitive affective construct, comprised of negative judgements of the self [30], which are global, undesirable, and characterised by an evaluation of the self as inherently weak, inadequate, or bad [31,32]. Shame is a subjective emotional response to negative events such as the making of mistakes, being wrong, and experiences of maltreatment [33,34]. Shame plays a central component in psychosocial functioning in its role as a trans-diagnostic emotion associated with many mental health conditions [35].

While shame and humiliation are often used interchangeably in literature [36], similarities and differences between these two constructs can be identified. Like shame, humiliation occupies space within the realm of "self-conscious emotions" [37]. These emotions are characterized by (a) a consciousness of the self and (b) some form of evaluation of the self. Both experiences require an individual to make an interpretation of an event as shameful or humiliating [38]. However, unlike shame, humiliation involves more emphasis on an interaction in which one is debased or forced into a degraded position by someone who is, at that moment,

more powerful [39]. Klein [38], in clarifying the distinction between shame and humiliation avers that: *"Shame is what one feels when one has failed to live up to one's ideals for what constitutes suitable behaviour in one's eyes as well as the eyes of others. Humiliation is what one feels when one is ridiculed, scorned, held in contempt, or otherwise disparaged for what one is rather than what one does"* [38]. Shame has been related to self-injurious behaviour [40], suicide attempts [41] and suicidal ideation [42]. Although previous systematic reviews have examined associations between many significant life events including, child sexual abuse [43–45], bullying [21–28] and self-harm or suicidality, the association between experiences of humiliation and/or shame and self-harm or suicidal behaviours among adolescents and young adults has not been well examined. Previous systematic reviews have examined the association of shame with many mental health conditions, including psychosis [31], anorexia and bulimia nervosa [32], depressive symptoms [33], and substance abuse [46]. Only one systematic review has studied the association between shame and self-harm in adults [47]. This indicated that individuals with a history of self-harm reported greater shame, and highlighted a correlation between shame and frequency of self-harm [47]. However, this latter work did not focus on young people, and centres on self-harm outcomes only. Therefore, the current systematic review is unique and the first of its kind, due to its primary focus on adolescents' and young adults' experience of shame and humiliation and associations with self-harm and suicidality.

The rationale for selecting two distinct age groups (adolescents and young adults) for this review is based on the increased prevalence of self-harm repetition among these age groups in recent years, with some interesting age and gender variations [48]. Whilst self-harm repetition rate is high among adolescent females (15–19 years old) in comparison to young adults; this repetition rate is also high among young adult males [48].

## Objectives

1. To systematically review studies that report the prevalence of humiliation and shame among adolescents and young adults with a history of self-harm and/or suicidal behaviours. 2. To systematically review studies that examine the association between humiliation and shame and self-harm and/or suicidal behaviours among adolescents and young adults.

## Methods

This systematic review protocol followed the Preferred Reporting Items for Systematic Reviews and Meta-Analysis Protocol guidelines (PRISMA-P) [49]. The protocol has been registered with International Prospective Register of Systematic Reviews (PROSPERO) [CRD42022289843].

*Eligibility Criteria*: All original studies published in the English language will be considered for this review. Peer reviewed articles and grey literature will be included.

Specifically, the following eligibility criteria will be fulfilled:

*Study Design*: Quantitative research studies including cross-sectional, prospective or longitudinal, and case control studies will be included. Mixed method studies will be included if there are quantitative measurements included in the study variables. Experimental studies/ quasi-experimental will be included if there are sufficient baseline data available. Qualitative studies, case reports, and case series will be excluded.

*Participants*: Adolescent or young adults (13–24 years of age).

*Exposure and Outcome*: Studies reporting the prevalence or association of humiliation or shame with self-harm, suicidal ideation, suicide attempts, and completed suicide will be included. Humiliation and/or shame measured by standard instruments or self-reported questionnaire or measured by single item/questions will be included.

*Setting*: No restriction by type of setting. Furthermore, there are no restrictions on the date of publications.

## Information sources

Electronic databases including MEDLINE, Web of Science Core Collection, CINAHL, PsycINFO, and Embase will be systematically searched to identify potential studies. Google Scholar (as a secondary source) will be searched (first 200 articles) to identify if any potential studies have been left out. The combination of Embase, MEDLINE, Web of Science Core Collection, and Google Scholar performed best, achieving an overall recall of 98.3 and 100% recall in 72% of systematic reviews [50]. The thesis and dissertation database ProQuest will be searched to identify relevant studies in the grey literature. Additionally, the reference list of the retrieved articles and/or previous systematic reviews in this area will also be scanned to identify further potential studies. The literature search will be conducted from 20th September 2021 to 29th April 2022. The studies will be searched from inception to 29th April 2022.

## Search strategy

Based on our initial search in the electronic databases, the following MESH terms or key/index terms were identified. A combination of these terms (at least one term from each category) will be used for conducting the literature search.

Boolean operators such as 'AND', 'OR', 'NOT' will be used to maximise the penetration of terms searched, and appropriate "wild cards" will be employed to account for plurals, variations in databases, and spelling.

Category 1

*Population*: Adolescent (MESH), young adult (MESH), teen, teenage.

Category 2

*Exposure*: Humiliation, degradation, shame (MESH) or embarrassment (MESH), harassment, victimisation, abasement.

Category 3

*Outcome*: Self-injurious behaviour (MESH), suicide (MESH), suicide attempted (MESH), suicide completed (MESH), self-harm, intentional self-injury, deliberate self-harm, overdose, deliberate self-poisoning, non-suicidal self-injury, self-mutilation, suicidal thought, suicidal ideation, suicidal intent.

Details of the search strategy has been included as a supplementary file.

## Data management

The literature search results (including citations, abstracts and full text) will be uploaded to Rayyan, an open source for the management of records for systematic reviews, where duplicates will be removed.

## Study selection process

Two authors (AMcL & AS) will independently screen the titles and abstracts yielded by the search against the inclusion criteria through Rayyan. We will obtain full reports for all titles that appear to meet the inclusion criteria, or where there is any uncertainty. Two of the review authors (AMcL & AS) will then screen the full text reports and decide whether these meet the inclusion criteria. We will seek additional information from study authors where necessary to resolve questions about eligibility. We will resolve disagreement through discussion. We will record the reasons for excluding studies through Rayyan.

## Data collection process

The relevant study details will be populated onto a pre-prepared data extraction sheet on Microsoft Word. The self-prepared data extraction sheet will include: the author, year of publication, country of study, study setting, population and sample, study design, outcome variables or measures, and main findings. Additionally, data relevant to methodological quality appraisal will be extracted from all of the included studies. Data will be extracted by two independent review authors (AMcL & AS).

## Risk of bias assessment

The studies will be assessed for methodological quality using the Joanna Briggs Institute (JBI) critical appraisal tools for analytical cross sectional (eight-item) [51], cohort studies (eleven-item) [52] or Case Control Studies (ten-item) [53]. The items include assessment on sampling, study setting, measurement of exposure, condition and outcome, identification and management of confounding factors, appropriateness of the statistical methods, and three additional items for cohort studies (i.e. duration of follow-up, dropouts, and strategies to address incomplete follow-ups) Two additional items for case control studies will be included (appropriate match of the case and control subjects, and the criteria used for identification of cases and controls). Each item in the JBI appraisal tools is answered as Yes, No, Unclear or Not applicable. Two review authors (AMcL & AS) will independently apply the tool to each included study, and record supporting information and justifications for judgements of risk of bias for each domain. Any discrepancies in judgements of risk of bias or justifications for judgements will be resolved by discussion to reach consensus between the two review authors, with a third review author acting as an arbiter if necessary.

## Data synthesis

The summary of the studies and risk of bias assessment results will be presented in tabular form in chronological order, starting from most recent. A descriptive summary of the findings including proportion studies reporting high versus low prevalence or significant versus non-significant association between humiliation or shame with self-harm and/or various types of suicidal behaviours will be presented. The studies reporting an association versus no association will be compared with the findings on risk of bias assessment. Furthermore, the studies reporting an association versus no association will be compared with sample size, study setting, study design, duration/frequency of outcome, and quality of measurement used across the studies.

If the studies are sufficiently homogenous in terms of population subtypes (clinical vs non-clinical and/or adolescents vs young adults), exposure (humiliation/shame), and outcomes (self-harm or suicidal behaviours); the results will be pooled for a meta-analysis. Cochrane RevMan 5 software will be used for meta-analysis. The dichotomous data will be presented as relative risks with 95% confidence intervals (CI). Continuous data will be presented as mean differences or standardised mean differences (Cohen's d) with 95% CIs (if the study outcome is measured by different scales). As the individual studies included cannot be expected to come from the same population of studies, pooled mean effect size will be calculated using the random effects model. Funnel plots and Egger's test will be used to determine publication bias. Heterogeneity will be assessed by visual inspection of forest plots, Cochrane's Q, and Higgins' test (I2). Moderation analyses may be conducted to understand the potential association with the effect sizes. For instance, to analyse the potential moderating effects of sex on humiliation or shame, the effect size of studies that consisted of females will be compared with the effect sizes of the studies that consisted of males.

If possible, sub-group analyses will be conducted according to the population subtypes (clinical vs non-clinical and/or adolescents vs young adults). If there is sufficient evidence to make recommendations, we will rate the certainty of evidence based on Cochrane methods and in accordance with the Grading of Recommendations Assessment, Development and Evaluation (GRADE) [54]. The GRADEpro will be used to construct a Summary of Findings. Accordingly, the certainty of evidence of the outcomes will be graded as high, moderate, low, and very low based on the five domains including; risk of bias, imprecision, inconsistency, indirectness, and publication bias.

## Discussion

To our knowledge, this is the first review to synthesise evidence on the prevalence of, and associations between the experiences of humiliation and shame and self-harm and suicidal behaviours among adolescents and young adults. As there is growing evidence on increased self-harm among this age group, it is important to identify population-specific risk factors for self-harm, suicidal ideation and behaviours which will have significance in formulating targeted and effective treatment and therapeutic services for adolescents and young adults. Identifying risk factors for self-harm and suicide during adolescence is key to achieving global health goals [55]. For instance, the review addresses UN Sustainable Development Goal 3, Target 3.4, to reduce by one third premature mortality from non-communicable diseases, including suicide, through prevention and treatment and promotion of mental health and well-being by 2030. This review is also aligned with the WHO Comprehensive Mental Health Action Plan 2013–2030. Concurrently, it is envisaged that this review will contribute to the evidence-base on factors influencing progression towards self-harm and suicide among adolescents and young adults.

## Supporting information

**S1 Checklist.**
(DOC)

**S1 File.**
(DOCX)

## Author Contributions

**Conceptualization:** Aoibheann McLoughlin, Anvar Sadath, Elaine McMahon, Katerina Kavalidou, Kevin Malone.

**Funding acquisition:** Aoibheann McLoughlin.

**Investigation:** Aoibheann McLoughlin.

**Methodology:** Aoibheann McLoughlin, Anvar Sadath, Elaine McMahon, Katerina Kavalidou.

**Project administration:** Aoibheann McLoughlin, Anvar Sadath, Kevin Malone.

**Writing – original draft:** Aoibheann McLoughlin, Anvar Sadath, Elaine McMahon, Katerina Kavalidou, Kevin Malone.

**Writing – review & editing:** Aoibheann McLoughlin, Anvar Sadath, Elaine McMahon, Katerina Kavalidou, Kevin Malone.

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
