## [Decision Letter · Decision Letter 0]

23 Aug 2022

PONE-D-22-08877Associations between Humiliation, Shame, Self-Harm and Suicidal Behaviours among Adolescents and Young Adults: A Systematic Review ProtocolPLOS ONE

Dear Dr. Vakkayil,

Thank you for submitting your manuscript to PLOS ONE. After careful consideration, we feel that it has merit but does not fully meet PLOS ONE’s publication criteria as it currently stands. Therefore, we invite you to submit a revised version of the manuscript that addresses the points raised during the review process.

This is an well written protocol covering one of the key aspect of suicidal behaviour. In addition of the Reviewer-1 comments, I am sharing my thoughts with a hope that it will increase the quality of the protocol and readability.

- Is it not the obj 2 should come first?

- Please consider providing a table containing the search command using Boolean operators.

- Since authors mention to run moderation analysis, please kindly consider stating what potential moderators authors are expecting to be a potential moderator/s. It may be my lack of understanding that I am not clear about what authors mean by ".......considering population characteristics and methodological aspects of studies". How does a methodological approach is relevant to the moderation analysis? Plus what sub-group analysis author are expecting to run? and what would be the population sub-type? A little description may increase my understanding.

- Please kindly consider describing the process of GRADE.

- Do the authors plan for present the meta-analysis by adolescent and young adults since they are two distinct age groups? Plus their experience/ understanding of Humiliation, Shame, way of Self-Harm etc. might be much different. I don't see any rationale of using such two distinct age groups in the introduction except to see that ".....adolescents and young adults has not been well examined".

We look forward to receiving your revised manuscript.

Kind regards,

Fakir Md Yunus, PhD, MSC, MPH, MBBS

Academic Editor

PLOS ONE

Journal Requirements:

"This study is being conducted as part of a funded collaborative study grant awarded by the Health Service Executive (HSE) - National Office for Suicide Prevention (NOSP), Ireland."

"This study is being conducted as part of a funded collaborative study grant awarded by the Health Service Executive (HSE) - National Office for Suicide Prevention (NOSP), Ireland.

The funders had and will not have a role in study design, data collection and analysis, decision to publish, or preparation of the manuscript."

Additional Editor Comments (if provided):

Reviewers' comments:

Reviewer's Responses to Questions

**Comments to the Author**

1. Does the manuscript provide a valid rationale for the proposed study, with clearly identified and justified research questions?

Reviewer #1: Yes

Reviewer #2: Yes

2. Is the protocol technically sound and planned in a manner that will lead to a meaningful outcome and allow testing the stated hypotheses?

Reviewer #1: Yes

Reviewer #2: Yes

3. Is the methodology feasible and described in sufficient detail to allow the work to be replicable?

Reviewer #1: No

Reviewer #2: Yes

4. Have the authors described where all data underlying the findings will be made available when the study is complete?

Reviewer #1: Yes

Reviewer #2: Yes

5. Is the manuscript presented in an intelligible fashion and written in standard English?

Reviewer #1: Yes

Reviewer #2: Yes

6. Review Comments to the Author

You may also provide optional suggestions and comments to authors that they might find helpful in planning their study.

Reviewer #1: I appreciate the effort of the review authors. I found the protocol very well written. The introduction and discussion sections are excellent. Please find my comments in the methods section.

1. Methods: Eligibility criteria (Study design): “Qualitative studies including case reports or case series will be excluded.” – Please rephrase as “Qualitative studies, case reports, and case series will be excluded.”

2. Information sources: “The literature search will be conducted from 20th September 2021 to 29 th April 2022.” – This is the search timing I guess. Please mention the search timeline/ period (for example: from inception till April 2022, or from 2000 to April 2022)

3. Search strategy: “Based on our initial scoping review via electronic databases, the following MESH terms or key/index terms were identified.” – What do you mean by scoping review? As a scoping review is a specific type of systematic review, it’s creating confusion. Did the author mean “initial exploration”?

4. Search strategy: As per the PRISMA 2020 guidelines, a comprehensive search strategy for all databases should be provided (at least as a supplementary file). Please provide a comprehensive search strategy for all databases.

5. Full-text screening: Please mention that the full-text screening will be conducted by two independent review authors.

6. Will the data be extracted by two independent review authors? If yes, please mention it. If not, please explain.

7. Risk of bias assessment: “The studies will be assessed for methodological quality using the Joanna Briggs Institute (JBI) critical appraisal tools for analytical cross-sectional (eight-item) [50] and cohort studies (eleven-item) [51]” – Previously authors have mentioned that they will include cross-sectional studies, case-control studies, cohort studies, experimental/ quasi-experimental studies (if they mention baseline data). Now, why only two JBI checklists have been mentioned? JBI has got separate checklists for separate study designs. Mention all as applicable.

Reviewer #2: This is a well described protocol with a solid rationale, and I believe the work will make a valuable contribution to knowledge. I look forward to seeing the results of the review when it is complete.

7. PLOS authors have the option to publish the peer review history of their article (what does this mean?). If published, this will include your full peer review and any attached files.

Reviewer #1: **Yes: **K.M. Saif-Ur-Rahman

Reviewer #2: No

---

## [Author Response · Author response to Decision Letter 0]

22 Sep 2022

Is it not the obj 2 should come first?

Our response: We agree with the suggestion. Objective 2 moved into the place of objective 1

Please consider providing a table containing the search command using Boolean operators

Our response: Details of the search strategy has now been included as a supplementary file

- Since authors mention to run moderation analysis, please kindly consider stating what potential moderators authors are expecting to be a potential moderator/s. It may be my lack of understanding that I am not clear about what authors mean by ".......considering population characteristics and methodological aspects of studies". How does a methodological approach is relevant to the moderation analysis? Plus what sub-group analysis author are expecting to run? and what would be the population sub-type? A little description may increase my understanding.

Our response: We revised the text and provided more clarity on moderation analysis and subgroup analysis

Please kindly consider describing the process of GRADE

Our response: We have included more details on the process of GRADE

Do the authors plan for present the meta-analysis by adolescent and young adults since they are two distinct age groups? Plus their experience/ understanding of Humiliation, Shame, way of Self-Harm etc. might be much different. I don't see any rationale of using such two distinct age groups in the introduction except to see that ".....adolescents and young adults has not been well examined".

Our response: We have included one paragraph to discuss the rationale for including the two distinct age groups. 

There is an absence of studies published in relation to humiliation and shame and the suicidal process in children. As such, it has been necessary to extend our review to incorporate young adults. We acknowledge the difference in the humiliation or shame experience across each distinct group and reference this in our review. 

In addition, this subgroup variation will be taken into account while analysing and interpreting the results. 

I appreciate the effort of the review authors. I found the protocol very well written. The introduction and discussion sections are excellent. Please find my comments in the methods section.

Methods: Eligibility criteria (Study design): Qualitative studies including case reports or case series will be excluded.” – Please rephrase as “Qualitative studies, case reports, and case series will be excluded.”

Our response: Thank you. We have rephrased the sentence as per reviewer suggestion

Information sources: “The literature search will be conducted from 20th September 2021 to 29th April 2022.” – This is the search timing I guess. Please mention the search timeline/ period (for example: from inception till April 2022, or from 2000 to April 2022”)

Our response: We have included the details of the search timeline

Search strategy: “Based on our initial scoping review via electronic databases, the following MESH terms or key/index terms were identified.” – What do you mean by scoping review? As a scoping review is a specific type of systematic review, it’s creating confusion. Did the author mean “initial exploration”?

Our response: We have rephrased the sentence

Search strategy: As per the PRISMA 2020 guidelines, a comprehensive search strategy for all databases should be provided (at least as a supplementary file). Please provide a comprehensive search strategy for all databases.

Our response: Yes, we have included the search strategy as a supplementary file 

Full-text screening: Please mention that the full-text screening will be conducted by two independent review authors

Our response: Yes, we have mentioned these details now

Will the data be extracted by two independent review authors? If yes, please mention it. If not, please explain

Our response: Yes. We have mentioned these details now. 

Risk of bias assessment: “The studies will be assessed for methodological quality using the Joanna Briggs Institute (JBI) critical appraisal tools for analytical cross-sectional (eight-item) [50] and cohort studies (eleven-item) [51]” – Previously authors have mentioned that they will include cross-sectional studies, case-control studies, cohort studies, experimental/ quasi-experimental studies (if they mention baseline data). Now, why only two JBI checklists have been mentioned? JBI has got separate checklists for separate study designs. Mention all as applicable.

Our response: We have included the details of JBI-Case control appraisal tool. 

Since the baseline data only be collected from the experimental or quasi experimental studies, we do not anticipate using a separate risk assessment tool for this category of studies. We will use the JBI - cross sectional critical appraisal. 

Reviewer #2: This is a well described protocol with a solid rationale, and I believe the work will make a valuable contribution to knowledge. I look forward to seeing the results of the review when it is complete.

Our response: Thank you

---

## [Decision Letter · Decision Letter 1]

10 Nov 2022

Associations between Humiliation, Shame, Self-Harm and Suicidal Behaviours among Adolescents and Young Adults: A Systematic Review Protocol

PONE-D-22-08877R1

Dear Dr. Vakkayil,

We’re pleased to inform you that your manuscript has been judged scientifically suitable for publication and will be formally accepted for publication once it meets all outstanding technical requirements.

Kind regards,

Fakir Md Yunus, PhD, MSC, MPH, MBBS

Academic Editor

PLOS ONE

Additional Editor Comments (optional):

Reviewers' comments:

Reviewer's Responses to Questions

**Comments to the Author**

1. Does the manuscript provide a valid rationale for the proposed study, with clearly identified and justified research questions?

Reviewer #1: Yes

Reviewer #2: Yes

2. Is the protocol technically sound and planned in a manner that will lead to a meaningful outcome and allow testing the stated hypotheses?

Reviewer #1: Yes

Reviewer #2: Yes

3. Is the methodology feasible and described in sufficient detail to allow the work to be replicable?

Reviewer #1: Yes

Reviewer #2: Yes

4. Have the authors described where all data underlying the findings will be made available when the study is complete?

Reviewer #1: Yes

Reviewer #2: Yes

5. Is the manuscript presented in an intelligible fashion and written in standard English?

Reviewer #1: Yes

Reviewer #2: Yes

6. Review Comments to the Author

You may also provide optional suggestions and comments to authors that they might find helpful in planning their study.

Reviewer #1: Thanks for addressing my comments and revising the manuscript. I don't have any additional comments.

Reviewer #2: I look forward to seeing the results of this work. This initial manuscript will be, I believe, appreciated by scholars in the field.

7. PLOS authors have the option to publish the peer review history of their article (what does this mean?). If published, this will include your full peer review and any attached files.

Reviewer #1: **Yes: **KM Saif-Ur-Rahman

Reviewer #2: No

<quillbot-extension-portal></quillbot-extension-portal>

---

## [Editor Report · Acceptance letter]

14 Nov 2022

PONE-D-22-08877R1 

Associations between Humiliation, Shame, Self-Harm and Suicidal Behaviours among Adolescents and Young Adults: A Systematic Review Protocol 

Dear Dr. Sadath:

I'm pleased to inform you that your manuscript has been deemed suitable for publication in PLOS ONE. Congratulations! Your manuscript is now with our production department. 

Kind regards, 

on behalf of

Dr. Fakir Md Yunus 

Academic Editor

PLOS ONE